# The Autocatalytic Chemical Reaction of a Soluble Biopolymer Derived from Municipal Biowaste

**DOI:** 10.3390/molecules29020485

**Published:** 2024-01-18

**Authors:** Elio Padoan, Enzo Montoneri, Andrea Baglieri, Francesco Contillo, Matteo Francavilla, Michéle Negre

**Affiliations:** 1Dipartimento di Scienze Agrarie, Forestali e Alimentari, Università di Torino, 10095 Grugliasco, TO, Italy; elio.padoan@unito.it (E.P.); negre.michele@gmail.com (M.N.); 2Dipartimento di Scienze delle Produzioni Agrarie e Alimentari, Università di Catania, 95123 Catania, CT, Italy; abaglie@unict.it; 3STAR Integrated Research Unit, Università di Foggia, 71121 Foggia, FG, Italy; francesco.contillo@unifg.it (F.C.); matteo.francavilla@unifg.it (M.F.)

**Keywords:** oxidation, autocatalysis, municipal biowaste, biosurfactants, biopolymers, antifungal agents, anaerobic fermentation, composts

## Abstract

The paper discusses the perspectives of further implementation of the autocatalytic properties of a soluble biopolymer (SBP) derived from municipal biowastes for the realisation of a biorefinery producing value-added bio-products for consumer use. The reaction of an SBP and water is reported to cause the depolymerisation and oxidation of the pristine SBP organic matter with the formation of carboxyl-functionalised polymers having lower molecular weight and CO_2_. These findings demonstrate the oxidation of the SBP via water, which could only occur through the production of O and OH radicals catalysed by the SBP. According to the adopted experimental plan, the anaerobic digestate supplied by an Italian municipal biowaste treatment plant was hydrolysed in pH 13 water at 60 °C. The dry product was re-dissolved in plain water at pH 10 and used as a control against the same solution with hydrogen peroxide at 0.1–3 H_2_O_2_ moles per SBP carbon mole added. The control and test solutions were kept at room temperature, in the dark or in a climatic chamber under irradiation with simulated solar light, until the pH of the solutions remained constant. Afterwards, the solutions were processed to recover and analyse the crude soluble products. The present work reports the results obtained for the control solution and for the test solutions treated in the presence and absence of H_2_O_2_, with and without pH control, in the dark and under irradiation with simulated solar light.

## 1. Introduction

### 1.1. Processes and Products for the Valorisation of Municipal Biowaste as Feedstock

Municipal biowaste (MBW) has been proposed as sustainable feedstock for the production of value-added water soluble biopolymers (SBPs) to use in different sectors of the chemical industry and agriculture in place of synthetic products obtained from fossil sources [1]. Since 2004, authors at the University of Torino and co-workers from other public and private institutions have published over 100 papers proving the environmental and economic sustainability and performance of SBPs as soil fertilisers, biostimulants for plant growth and crop production, anti-pathogen agrochemicals for the cultivation and protection of plants, diet supplement for animal husbandry, templates for the fabrication of nanocrystalline materials, photosensitisers for the photoremediation of industrial wastes containing organic pollutants, biopolymers for the manufacture of plastic materials, biosurfactants to use in the formulation of detergents, auxiliaries for textile dyeing and emulsifiers, and for the remediation of soil contaminated by organics and metals of dismissed industrial sites [1]. The multiple uses of SBPs are possible thanks to the products’ water solubility properties and organo-mineral composition, which comprise macromolecules, made from aliphatic and aromatic C chains, and acidic and basic functional groups bonded to a variety of mineral elements [2,3]. 

For the production of SBPs, an integrated chemical–biochemical approach has been used, which includes the chemical hydrolysis and ozonisation of the as-collected MBW [2] or fermented MBW under anaerobic and aerobic conditions [1]. The results of these studies have encouraged the belief that a conventional MBW–biochemical treatment plant, integrated with the proper facilities to carry on the chemical hydrolysis and oxidation processes, could be converted into a biorefinery producing conventional biogas and/or compost [4,5,6], together with the above multipurpose multifunctional SBPs. In the authors’ view, it was necessary to optimise the SBP’s performance in some applications for the realisation of the envisioned biorefinery to be competitive with fossil-based refineries.

A typical example needing optimisation is the SBP product obtained via hydrolysis from the MBW anaerobic digestate. This product contains organic and mineral matter. The organic matter is constituted by a pool of molecules with widely different molecular weight, ranging from 5 to over 750 kDa, and with different chemical compositions [1]. They are made from aliphatic C chains and aromatic lignin-like C moieties functionalised with acidic and basic groups of different strength, which are bonded to a wide range of mineral elements, such as Na, K, Ca, Mg, Al, Fe, Si in 0.2–9 *w*/*w* % content and Cu, Ni, Zn, in 20–180 ppm content [3]. These features are memories of the chemical moieties and macro-molecularity of the pristine MBW polysaccharide, protein and lignocellulosic proximates [2] from which the SBP is derived. 

Thanks to the above chemical composition, the SBP has been found to have good potential to perform as a surfactant and biopolymer to manufacture plastic articles. Studies assessing the performance of the SBP in real operational conditions have shown that the product’s black colour is a drawback for the manufacture of finished formulates to be used as fabric detergents or auxiliaries for fabric dyeing [7]. A good property for these applications is the SBP’s behaviour in water solutions, capable of forming large micellar aggregates sequestering the dirt from the fabric and transferring it to the aqueous washing medium. By the same behaviour, in the case of textile dyeing, the SBP allows one to control the release of the dye from the dyeing bath to the fabric to achieve colour uniformity. The drawback is the colour memory of the SBP remaining in the treated fabric in form of a yellow dark–light nuance [1]. This makes the SBP less competitive with commercial surfactants for specific uses. Also, the SBP’s performance regarding the manufacture of plastic articles is severely limited by its lack of film-forming properties and the poor mechanical properties [8]. All these drawbacks stem from the content of aromatic lignin-like moieties in SBP, contributing a black colour [7], and the mechanical rigidity and lack of plasticity of the obtainable plastic sheets [8]. Further studies have attempted to reduce the content of aromatic C in the SBP and bleach it via ozonisation [7]. The results have shown that this is possible, but ozonisation causes severe depolymerisation [2,7]. The reaction yields 30% of products with a molecular weight ≥ 100 kDa and 70% of products with a molecular weight < 5 kDa, the latter ones characterised by the complete loss of the surfactant properties of the pristine SBP. 

### 1.2. The Stake of Further Research for the Valorisation of Municipal Biowaste as Feedstock

The multiple applications of SBPs allow us to envision a potential MBW-based biorefinery producing new value-added multipurpose chemical specialities. At present, there are no commercial chemical products made from MBW, except for biogas, anaerobic digestate and composts [1]. A typical example is the compost made from food waste anaerobic digestate mixed with urban gardening residues produced by the Italian Acea MBW treatment plant [9]. This product, however, has a very low commercial value of ≤30 EUR/t [1,10]. Our previous work on SBPs [1] proves that exploitation of MBW as feedstock can generate new relevant benefits for waste management and the biobased economy. In Europe, about 100 Mt/yr of MBW is generated by 750 million EU citizens and treated by 18,000 plants [11,12]. About 35 Mt/yr of MBW is processed by anaerobic and aerobic fermentations, yielding biogas, anaerobic digestate and/or compost [13]. The rest is landfilled or incinerated emitting 11,000 Mm^3^/yr of CO_2_, 14,000 Mm^3^/yr of CH_4_, 2–4 kt/yr of dust and 154 kt/yr of other GHG and toxic organics [14]. On the other hand, the EU chemical industry produces 330 Mt/yr^1^ of organic chemicals from fossils to make plastics and chemical products [15,16,17] generating 1037 Mt/yr of CO_2_ emission. Sale prices [18,19,20,21,22] for plastics and high performance surfactants run 1500–150,000 EUR/t, worth a total market value of 84–8400 billion EUR/yr. By comparison, the turnover of biobased plastics and chemicals [16,17] is 70 billion EUR/yr. In this scenario, improvements in the yield and quality of SBPs could, in principle, allow one to obtain new biowaste-based commercial products, which are competitive with current commercial products and could replace large portions of EU chemicals obtained from fossil feedstock, thus increasing the market output of biobased chemicals and materials.

The realisation of these perspectives could generate considerable environmental, economic and social benefits. For examples, current MBW treatment plants are not cost effective [1]. The value of the produced biogas and compost is not enough to cover the collection and processing costs of the biowastes. Eighty per cent of it is paid for by citizens’ taxes. The conversion of the MBW treatment plants into biorefineries, producing the improved SBPs competitive with current commercial products, would allow one to raise the MBW plants’ net revenue by two–three orders of magnitude. This fact, in turn, would allow one to obtain a number of effects, i.e., reducing citizens’ taxes for waste disposal; establishing a healthier living environment due to the reduction in GHG emissions from landfill sites and from the fossil-based chemical industry; generating new jobs in biowaste-based refineries; convincing citizens that wastes are a source of economic, social and environmental benefits rather than a source of high costs; promoting social acceptance of the new biorefineries; improving citizens’ attitudes towards waste reuse and recycling; contributing to the standardisation/certification of biobased products; and updating the EU REACH chemicals policy by including SBPs as authorised chemical specialities for the chemical market.

### 1.3. The Autocatalysis Concept

Based on the drawbacks (Section 1.1) and the potential benefits (Section 1.2) of SBPs, the authors have conceptualised MBW autocatalytic chemical reactions as a mean to improve the yield of the production process and the properties of the SBP. According to the authors’ vision, the oxidation reactions of the MBW-derived SBPs might occur at room temperature in a water solution without added chemical reagents and/or catalyst. In line with the current trend to develop green chemical processes [23], the envisioned oxidation reactions employing the waste-derived SBP as substrates and catalysts and water as the greenest solvent and oxidant would not require the consumption of energy and reagents and the synthesis, recovery and/or regeneration of expensive catalysts. Thus, the implementation of SBP autocatalytic reactions on an industrial level would constitute an example of the most desirable, eco-friendly and cheapest process.

The assumption that the SBP could perform as a substrate and catalyst is supported by previous findings. The SBP, thanks to its acidic and basic functional groups, has been shown to bond with Fe ions, well-known catalysts for Fenton-like reactions performed in advanced oxidation processes for water remediation [24,25]. Indeed, other SBPs (e.g., the green compost-derived SBP) have been proven to perform as photosensitisers [26] in advanced photocatalytic oxidation processes [24] and to promote the oxidation of many organic pollutants present in industrial wastewaters under irradiation with simulated solar light. Also, depending on the concentration in water, SBPs have been found to promote their auto-oxidation [26]. The same products have been shown to catalyse the oxidation of ammonia to nitrogen [27], even in the absence of irradiation. These findings suggest that, in addition to their photosensitising properties, the SBPs could perform also as catalysts for their mild auto-oxidation under proper experimental conditions. A somewhat analogous virtuous waste cycle is reported in a doctoral thesis [28] describing functionalised products derived from agro-industrial biomass waste to catalyse a number of reactions applied to biomass for the production of platform chemicals. 

The valorisation of fermented MBW materials as feedstock for the production of other value-added products than soil amendments/composts is highly worthwhile. On one hand, the use of MBW anaerobic digestates as soil amendments is restricted due to their ammonia content and disadvantaged by the high cost of the secondary treatment to make the digestate comply with the current legislation [27]. On the other hand, the market value of the compost is very low [1,10] and does not cover the compost production cost. The potential market value of the SBPs obtainable from the MBW anaerobic digestates and composts would allow one to increase by two–three orders of magnitude the net revenue of the current conventional biochemical MBW treatment plants upon integrating them with the chemical facilities allowing the production and commercialisation of SBPs [1]. As a contribution to the realisation of these perspectives, the present work was undertaken to investigate the behaviour of the aqueous solutions containing the SBP derived from the MBW anaerobic digestate supplied by the conventional MBW-treatment plant located in Italy [9]. 

According to the adopted experimental plan in the present work, the behaviour of the SBP solution without added reagents and/or catalysts, irradiated and not irradiated with simulated solar light, was investigated in comparison with the SBP solution containing added alkali and/or hydrogen peroxide. The complex chemical composition of the SBP [1,3] offered ground supporting the foreseen autocatalysis. On the other hand, it made the demonstration of the autocatalysis concept quite challenging and required the application of an analytical protocol specifically developed [1]. This included fractionation of the tested SBP via sequential ultrafiltration with membranes with different cutoffs in the 0.2–750 kDa range, determination of the content of organic and inorganic matter, and of elemental C and N, characterisation of functional groups by solid-state NMR spectroscopy coupled with wet chemical analyses, and mathematical elaboration of the experimental data. The results reported here in after will show how this analytical protocol is particularly necessary in the case of the present work aiming to assess the autocatalytic effect of the tested SBP derived from the MBW anaerobic digestate. 

## 2. Results

### 2.1. Treatments of SBP and Recoveries of C in Crude Soluble Products

According to the experimental conditions reported in Section 4.1, the SBP available from previous work [1] was dissolved in plain water at pH 10. The solution was used as a control against the same solution added with hydrogen peroxide. The purpose was to test the effect of the added hydrogen peroxide oxidant in comparison with water as the only terminal oxidant in the control solution. Table 1 reports the experimental conditions under which the SBP solution was treated and also the soluble carbon as mol/mol % of the total C in the SBP before the treatments, which was recovered from the control solution, and the solutions with added hydrogen peroxide at the end of the treatments. During the treatments, some insoluble material formed. This was separated from the soluble phase via centrifugation. The recovered insoluble material accounted for 3–9% of the total C in the SBP before the treatments (Appendix A). The formation of insoluble matter in the oxidation of lignocellulose materials has been reported also in the ozonisation of SBP [7] and of pine kraft lignin in alkaline solution [29]. It was attributed to the dehydrogenative coupling and cross-linking of ozonised phenolic moieties induced by active oxygen radicals. In the present work, the insoluble co-product was undesirable. Values of the recovered C mol/mol % for the recovered soluble material (Table 1) and for the insoluble material were calculated from the respective recovered dry mass weights and C contents reported in Appendix A. 

The data in Table 1 for treatments No. 1-D0 and No. 4-L0, respectively performed without and with light irradiation, in the absence of added H_2_O_2_ show that the pH of the starting SBP solution in water decreases after 14 days from 10 to 9. For treatments No. 2-D2, 3-D3, 5-L2, 6-L3, performed in the presence of added H_2_O_2_ at a 2–3 H_2_O_2_/C mole ratio, without and with light irradiation, the pH decreases to about 5. According to previous works on the ozonisation of the as-collected MBW [2] and SBPs obtained from fermented MBW [7], the pH decrease reported in Table 1 is consistent with the oxidation of SBP organic matter and the formation of carboxylic functional groups and/or CO_2_. The same treatments were replicated adding KOH to keep the pH 10 of the pristine SBP solution constant during the treatments performed under the same above conditions. The data for treatments No. 7–12 in Table 1 show that the amount of added KOH needed to maintain pH 10 is equivalent to the formation of 0.24 acid equivalents produced per mole of the starting organic carbon (H^+^/C eq/mol) in the case of the No. 7-D0 and No. 10-L0 treatments, of 0.34 H^+^/C eq/mol in the case of No. 8-D2 and No. 11-L2, and 0.40 H^+^/C eq/mol in the case of the No. 9-D3 and No. 12-L3 treatments. The data confirm that the production of organic functional COOH groups and/or CO_2_ occurs also for the SBP solutions containing no added H_2_O_2_ and that the produced H^+^/C eq/mol amount depends on the amount of added H_2_O_2_. 

Parallel to the pH decrease, Table 1 shows that the recovered soluble C at the end of treatments No.1–6 decreased along with a decrease in pH of the recovered solution. Similar trends were observed for the weight of the recovered soluble matter and for the C/N mol/mol ratio in the dried recovered soluble matter (Appendix A). 

An evaluation of the data for the recovered soluble C in Table 1 should account for the variability of the measured values, due to handling the recovery of the reaction medium, separating the soluble and insoluble phases, drying, weighing and analysing the products for their C and N content (see Section 4). To this purpose, Table 2 reports the mean values and standard deviations calculated over the values for treatments No. 1–6 and No. 7–12, separately. Compared to treatments No. 7–12, treatments No. 1–6 are characterised by significantly lower mean values and also much higher standard deviations values. The large differences in mean and standard deviation values between the two groups stem mainly from the production of CO_2_ and its fate. While in treatments No. 7–12 at a constant pH of 10, the produced CO_2_ remains in the recovered material in the form of a carbonate, in treatments No. 1–6, it is lost in the gas phase. On one hand, the 5.9 standard deviation for treatments No. 7–12 is likely to represent largely variability due to handling, separating and analysing the crude matter at the end of the treatments. On the other hand, a comparison of the data for the two groups of treatments could potentially allow for an assessment of the effect of pH on the soluble organic matter recovered in the different treatments. This poses the issue of understanding how much of the recovered soluble C in treatments No.7–12 is due to the formation of CO_2_ and of organic COOH functional groups.

To assess the relative contributions of CO_2_ and organic C, samples of the products obtained in Table 1 treatments were further treated with HCl in order to obtain CO_2_-free samples (see Section 4). Figure 1 reports the composition of C types and functional groups in the products determined by 13 C NMR solid state spectroscopy. 

The content of the C types and functional groups listed in Figure 1 was estimated from the measured areas of the 13C NMR resonance band covering the following chemical shift (δ, ppm) ranges: 0–53 for aliphatic (Af) C; 53–63 ppm for amine (NR) and methoxy (OMe) C; 63–95 ppm for alkoxy (OR) C; 95–110 ppm for anomeric (OCO) C; 110–140 ppm for aromatic (Ph) C; 140–160 ppm for phenol/phenoxy C (PhOY, Y = H, R); and 160–185 ppm for carboxylate and amide (COX: X = OM, NR; M = metal, R = H, alkyl and/or aryl) C. The total integrated band area over the 0–185 ppm range was assumed to represent the total C moles in the analysed sample (see Section 4). 

For the crude soluble products listed in Figure 1A,B, the COX resonance signal was generally very broad, covering all the 160–185 ppm resonance range. Some spectra exhibited a high intensity sharp signal overriding the broad band in the 160–185 ppm range. In these cases, the measured band area for the COX resonance accounted for 40–60% of the total integrated band area over the 0–185 ppm range. This occurred particularly for the products obtained in the treatments at controlled pH 10 (Figure 1B), which were expected to contain potassium carbonate, formed as a consequence of the mineralisation of the SBP organic matter in the aqueous alkaline medium. Potassium carbonate is characterised by a sharp intense 13C resonance signal at 170.3 ppm [30]. To determine the content of potassium carbonate, some selected samples of the crude soluble products obtained at constant pH 10 were treated with HCl (see Section 4), and the expected CO_2_-free products were analysed via 13C NMR spectroscopy. The same HCl treatment and spectroscopic analysis was performed on selected samples of the crude soluble products obtained at an acidic pH. This allowed us to assess the possible effects of the HCl treatments, other than decarbonation, on the CO_2_-free organic matter of the pristine crude soluble products.

### 2.2. Recoveries of Soluble Organic C and CO_2_ for Treatments in Absence of Added H_2_O_2_

For the scope of the present work, it was of primary importance to compare the crude soluble products obtained in the treatments carried out in absence of added H_2_O_2_ with their corresponding samples treated with HCl. According to the data in Figure 1, the 1-D0 and 4-L0 samples do not show any significant composition difference compared to the corresponding 1-D0dec and 4-L0dec samples subjected to the HCl decarbonation treatment. For each C type and functional group, Table 3 reports the mean and standard deviation calculated for the 1-D0 and 1-D0dec samples and for the 4-L0 and 4-L0dec samples, separately. 

It may be observed that the relative standard deviation values range from 2.8% to 11%, except in the case of the PhOY functional group, which shows rather large 19% and 51% values. This poses doubts about the significance of the 13C signals measured in the PhOY 140–160 ppm. In most cases, the broad PhOY resonance band could hardly be distinguished from the background noise. 

On the contrary, the data in Figure 1 show evidence of large significant composition differences for the 7-D0 and 10-L0 samples compared to the corresponding 7-D0dec and 10-L0dec samples subjected to the HCl decarbonation treatment. In these cases, the relative standard deviations were found to range from 16% to 58% of the mean values calculated for all C types and functional groups. The data in Figure 1 show that the COX content, much lower in the decarbonated samples than in the pristine crude soluble 7-D0 and 10-L0 samples, is primarily responsible for the large composition differences measured for these pairs of samples. These results prove that the measured area of the broad COX resonance band in the 7-D0 and 10-L0 products includes contributions of the resonance signal of CO_2_ C (in form of potassium carbonate) and of organic carboxyl C (COXorg), which were calculated according to the following Equations (1) and (2):COXorg contribution % = A × B/C(1)
CO_2_ contribution % = D − COXorg contribution %(2)
where (Figure 1) A and C are the values of COX and Af, respectively, for 7-D0dec or 10-L0dec; B is the value of Af for 7-D0 or 7-L0; and D is the value of COX for 7-D0 or 7-L0.

Table 4 reports the calculated values of COXorg and CO_2_ (mol/mol %) carbon contained in the products obtained in the 1-D0, 4-L0, 7D0 and 10-L0 treatments listed in Table 1 and in Figure 1A,B. As expected, the data confirm that in the products obtained in treatments 7-D0 and 10-L0 at constant pH 10, the produced CO_2_ is retained in the form of potassium carbonate in the strong alkaline water phase. On the contrary, no potassium carbonate is found in the products obtained in treatments 1-D0 and 4-L0 without pH control. The amount of CO_2_ calculated from the 13C spectroscopic data for 7-D0 and 10-L0 products corresponds to the values calculated from the consumption of the alkali to keep pH 10 constant during the treatments (see Table 1, column H^+^/C). 

Similar calculations using Equations (1) and (2) were applied to the data reported in Figure 1B for the crude soluble product obtained in treatments 11-L2 and 11-L2dec at constant pH in the presence of hydrogen peroxide at a 2 H_2_O_2_/C mol/mol ratio. The calculated amount of CO_2_ C in the 11-L2 sample was 53.2 mol/mol % against 8.7 mol/mol % for organic COX C. These results indicated that the mineralisation of organic C in the presence of hydrogen peroxide is more than double than that (Table 4) calculated for the 7-D0 and 10 L-0 samples of the crude soluble product obtained in the treatments at constant pH 10 in the absence of hydrogen peroxide without and with light irradiation, respectively.

### 2.3. Molecular Weight Distribution in Pristine SBP and Crude Soluble Products Obtained in Absence of Added H_2_O_2_

Further information on the effects of the 1-D0, 4-L0, 7-D0 and 10-L0 treatments of SBP in the absence of added H_2_O_2_ was obtained by fractionating the recovered crude soluble materials through sequential membrane ultrafiltration. To this end, the pristine SBP and the recovered soluble products were fed to polysulphone membranes with decreasing molecular cutoffs at 750, 150, 100, 50, 20, 5, and 0.2 kDa, and the collected retentates at each step and the final permeate through the 0.2 kDa membrane were collected, weighed and analysed for their C content. Figure 2 reports the results of the fractionation process. 

The data in Figure 1A show that the R750, R150 and R100 in order of decreasing weight abundance are the major high molecular weight fractions of the pristine SBP solution (0-SBP in Figure 1) ultra-filtered readily after its preparation at pH 10. However, keeping the SBP in the alkaline aqueous solution for 14 days (treatment No. 1-D0 in Table 1) yields a product (1-D0 in Figure 1A) exhibiting a drastic compositional change with the R750 fraction reduced at a 5.2% level and the R150 and P0.2 fraction increased up to 54% and 40.8%, respectively, compared to the composition of the pristine SBP. For the SBP solution irradiated with simulated solar light for the same time, the crude soluble product (4-L0 in Figure 1A) exhibits a further reduction in the fractions’ molecular weight, with the R750 and R150 accounting for 6.5% and the R20 and P0.2 fractions accounting for over 89.1% of the total recovered material. As shown in Table 1, during the 14-day treatments, the pH of the two 1-D0 and 4-L0 solutions decreased to 9. For the products obtained in the treatments carried out at constant pH 10 (i.e., 7-D0 and 10-L0 in Figure 1A), the reductions in molecular weight seemed less compared to the 1-D0 and 4-L0 products in Figure 1A. For 7-D0 (in Figure 1A) obtained without irradiation of the pH 10 reaction medium, the R750 fraction accounted for 30% and the R150 for 6.5%, compared to 5% for R750 and 54% for R150 in the 1-D0 product (in Figure 1A) obtained without pH control. For 10-L0 (in Figure 1A) obtained from the irradiated reaction medium, the R50 accounted for 14.2% against 0% for the 4-L0 product, while the lower molecular weight fractions R20, R0.2 and P0.2 altogether were 86.2% in the 10-L0 product and lower than 93.3% in the 4-L0 product (Figure 1A) obtained without pH control of the reaction medium. For the 4-D0 and 10-L0 products, the P0.2 fraction is supposed to contain potassium carbonate, the product of the mineralisation of organic C (see Section 2.2). 

Figure 1B reports the total C distribution over each of the above materials. For the pristine SBP (0-SBP in Figure 1B), most of the total carbon (69%) is accounted by the R750 fraction. For the crude soluble products obtained in each treatment, the carbon recoveries are given as *w*/*w* % relative to the carbon in the pristine SBP. The C recovery values are calculated based on the mass data given in Figure 1A and the C content measured for each fraction. It may be observed that the pattern of C recovery distribution changes significantly depending upon the type of treatment. The carbon recovered with the R750 fractions of the crude soluble products ranges from 0.5% for 10-L0 to 30% for 7-D0, while most of the remaining C (46–81%) is recovered with the lower molecular weight fractions. 

Both the mass and C recovery data in Figure 1 indicate that all treatments cause depolymerisation of the pristine SBP organic matter and that the effect is stronger in the treatments carried out via irradiation and/or without pH control of the reaction medium. These findings offer highly relevant information, which, coupled with the results reported in Section 2.1 and Section 2.2, undoubtedly demonstrate the autocatalytic properties of SBP to react with water and lead to the depolymerisation and mineralisation of its own organic matter. 

### 2.4. Products Obtained in the Presence of Added H_2_O_2_

Figure 3A reports the C recoveries, as mol/mol % relative to the carbon of the pristine SBP, with each fraction isolated from each crude soluble product obtained in all SBP treatments listed in Table 1. The C recoveries are calculated from the weight and carbon content of the fraction isolated through the same membranes listed in Figure 2. The different colours of the histogram columns identify the different retentates and permeates obtained via the ultrafiltration. 

The data show the effect of hydrogen peroxide on the molecular weight distribution of the crude soluble products. For the treatments carried out at constant pH 10, without (7-D0, 8-D2, 9-D3) and with irradiation (10-L0, 11-L2, 12-L3), the data show an increase in the lowest molecular weight fractions R0.2 and/or P0.2 occurring in the presence of added H_2_O_2_ compared to the reaction in the absence of H_2_O_2_. This is readily evidenced in Figure 3B, which reports the total C recovered with the retentate (R0.2) and permeate (P0.2) fractions as the % of the total C recovered with all fractions.

It may be observed in Figure 3B that, in the case of the treatments carried out without light irradiation and with no pH control (1-D0, 2-D2 and 3-D3), the total production of R0.2 and P0.2 increases upon increasing the added content of H_2_O_2_ from 2 to 3 moles per SBP C mole. In the other cases, no effect or significant trend appears evident by increasing the content of H_2_O_2_ above 2 moles per SBP C mole. A somewhat similar trend is observed due to light irradiation. Compared to the 2-D2 treatment, the 5-L2 treatment caused a strong increase in the total production of R0.2 and P0.2. In all other cases, no definite effect or trend may be picked out as being caused by light irradiation. 

The data in Figure 3 show that the depolymerisation and/or mineralisation of SBP organic matter is particularly evident in treatments 3-D3, 5-L2, and 6-L3, where the sum of the R0.2 and P0.2 % values in Figure 3B accounts for 74–93% of the values reported in Table 1 for the total C recovered with 3-D3, 5-L2, and 6-L3 treatments. Considering the data in Table 1 and Figure 1, Figure 2 and Figure 3, it appears evident that the SBP treatments in the presence of hydrogen peroxide at the 2 and 3 H_2_O_2_/C ratio are too drastic, due to the enhancement of depolymerisation and mineralisation via the oxidation of the SBP organic matter. Similar results were obtained for the SBP treatments at controlled pH 10 and the 0.1–0.5 H_2_O_2_/C mole ratios not reported here. Under these conditions, 0.06–0.15 H^+^/C eq/mole were produced. This indicated that the oxidation of the SBP organic C to organic COOH functional groups and/or CO_2_ was quite less than it was in treatments No. 7–12 at the 2–3 H_2_O_2_/C mole ratio reported in Table 1.

### 2.5. Chemical Composition and Properties of the Molecular Weight Fractions Obtained in All Treatments

Figure 4 and Figure 5 report data related to the chemical composition and surface activity properties of the fractions obtained from the crude soluble products obtained in all treatments carried out in the present work. More detailed data are given in Appendix A.

For reactions involving products of complex chemical composition such as SBP, the ratio of the carbon to nitrogen (C/N) content in a product has been used as an index for the effect of a chemical or a biochemical reaction on the chemical nature of the product. For example, for the chemical hydrolysis of a wide variety of municipal biowaste composts, the following Equation (3) has been proposed [1]: C/N =1.81 + 2.60 Z(3)
where Z is the ratio of the measured total aliphatic, aromatic, carboxylic, phenol, phenoxide, methoxy, amine, amide and ketonic carbon over the sum of amine, amide and carboxylate functional groups. A similar significant linear relationship has been confirmed between the C/N values of different composts with the C/N values of the derived SBP products [27]. The relationship allows one to predict the chemical composition of SBPs obtained from different composts. According to the authors of [27], it constitutes a valuable tool to assist the industrialisation of the SBP production process.

The data in Figure 4 show that the C/N values vary over a very wide range, from a minimum 3.3 value for the P0.2 fraction of the crude soluble product obtained in the 3-D3 treatment, listed in Table 1, to 49 for P0.2 of the crude soluble product obtained in the 15-L0 treatment of the pristine SBP solution carried out at controlled pH 10, under irradiation and in the absence of added H_2_O_2_. Most of the fractions in Figure 4 have C/N values different from the 8.7 value for the pristine SBP reported in Appendix A. The variability of the C/N and molecular weight values of the fractions composing the crude soluble products obtained in the different treatments of the pristine SBP reflects the complexity of the supply chain that includes the pristine MBW and the SBP anaerobic digestate, from which the products are derived. In essence, each fraction in Figure 4, obtained via ultrafiltration of the crude soluble products described in Section 2.1, Section 2.2, Section 2.3 and Section 2.4, is a mixture of molecules with not only different molecular weights, but also different chemical compositions. With specific references to the P0.2 fractions, the C/N values higher than 8.7 (measured for the pristine SBP) are likely contributed mostly by the content of potassium carbonate CO_2_. This occurs for most of the P0.2 fractions isolated from the crude soluble products obtained in the treatments at constant pH 10, as anticipated by the data for the crude soluble products obtained at constant pH 10 (see Section 2.2). On the contrary, low C/N values, for example, C/N values of 3.3 and 4.4 for P0.2 isolated from the 3-D3 and 6-L3 crude soluble products, indicate the presence of small molecules containing N, as organic oxymes isolated in the ozonisation of SBP [7].

Table 5 reports mean C/N values calculated from the single C/N values (Figure 4 and Appendix A) of the fractions isolated as retentates or permeates through membranes with the same cutoff values. 

Mean C/N values have relatively high relative standard deviations, which do not allow for the confirmation that the differences between average values are statistically significant. Many factors contribute to high standard deviations. For the complex molecular mixtures dealt with in the present work, the major factors are the different reaction conditions adopted, e.g., the applied H_2_O_2_/C mole ratio, together with the replicability of the course of the reaction under the same experimental conditions and the change of the solution conformation of the molecular pool constituting the crude soluble products from which the corresponding fractions in Figure 4 are isolated. The polymeric molecules in the crude soluble products, through their basic and acidic functional groups, may establish intermolecular H-bonds. Depending on their concentration in water, these molecules form aggregates [24,25,26,27,28,29,30,31,32,33,34,35,36] of different sizes trough the H-bond network established between macromolecules of different compositions. In dilute solutions, the size of these aggregates [37] is likely to decrease, and therefore the composition of retentates through a membrane with a specific cutoff may change depending on the concentration of the solution fed to the membrane. Yet, within the limitations posed by the relatively high variability of the C/N parameter, the data in Table 5 show some apparent differences that may help to rationalise the nature of the crude soluble products. For example, the lowest molecular weight P0.2 fractions have the highest C/N average (14.4) and standard deviation (11.0) corresponding to STDr 76%. These results are most likely due to the fact that the P0.2 fractions contain variable amounts of carbonates, as anticipated in Section 2.2. The same may be true for the R0.2 fractions. By comparison, all other fractions exhibit lower average C/N values in the 6.7–9.5 range and lower STDr in the 20–55% range. Excluding the P0.2 and R0.2 fractions, the apparent order of average C/N values is R20 > R750 > R150 = R50 > R5. The C/N data therefore indicate that, based on Equation (3) and on the molecular weight, the R750 and R5 have, respectively, the lowest and the highest relative content of amino carboxylic and peptide functional groups plausibly organised in protein-like moieties with different molecular weights.

According to previous works [7,37,38], the measurement of surface tension in water solutions is a diagnostic tool, which indicates the behaviour of a product in a solution and its potential performance as surfactant. In the present case, surface tension measurements were carried out as a mean for rating the many different fractions listed in Figure 3 and Table 4 on the basis of the surface tension measured for their water solutions and therefore for their potential prospected value in the chemical market. The results of these measurements for selected samples are reported in Figure 5 and also in Appendix A. The graphical representation of the measured surface tension (γ) values in different colours (Figure 5) allows us to observe readily that the experimental γ data may be divided into three groups, i.e., γ < 50; 50 ≤ γ < 60; and γ ≥ 60. 

The data points and the standard deviation bars in Figure 5, and the statistical analysis given in Table 6, show that the mean values of the three groups are significantly different from each other. The SBP treatments at H_2_O_2_/C 0.5–0.1 mole ratios without pH control, and without or with irradiation yield products with the lowest γ < 50 values (Figure 5). All others have γ > 50 values. All samples obtained by virtue of the autocatalytic properties of SBP in the absence of added H_2_O_2_ seem to exhibit the highest γ ≥ 60 surface tension values. On the other hand, the treatments in the presence of hydrogen peroxide at 2–3 H_2_O_2_/C mol/mol seem to yield fractions with some improvement in the surface activity properties, but they produce a high degree of depolymerisation of the pristine SBP organic matter. The treatments at H_2_O_2_/C 0.5–0.1 mole ratios produce also depolymerisation of the pristine SBP organic matter but yield the fractions exhibiting the best surface activity property. 

## 3. Discussion

### 3.1. The Autocatalytic Property of SBP

SBP depolymerisation (Figure 2) and mineralisation (Figure 1) in water, induced by the 1-D0, 4-D0, 7-L0, 10-L0 SBP treatments in the absence of added H_2_O_2_, prove that the SBP auto-catalyses its own oxidation. In the absence of added oxidant reagents, the reaction can only occur through the generated oxygen and/or hydroxyl radicals [2,7] catalysed by SBP [1]. Therefore, under these conditions, water acts as a solvent and oxidizing agent. The data also demonstrate that the SBP auto-oxidation occurs both with and without light irradiation of the SBO solution. 

The auto-oxidation of a compost-derived SBP in a water solution at 500 mg L^−1^ under irradiation with simulated solar for 0–15 h has been reported [1] to yield crude soluble products with an increasing content of organic carboxylic functional groups and to cause a decrease in the SBP aromatic chromophores and fluorophores, a lighter colour and a complete loss of the surfactant properties of the pristine compost-derived SBP. Photo-bleaching, as evidenced by the lighter colour of the product, was more marked in the presence of added hydrogen peroxide. The auto-oxidation of the SBP solution without irradiation and with simulated solar light, as observed for the SBP tested in the present work, is so far unknown in the literature.

Photo-assisted chemical reactions under light and dark conditions are known to occur for systems containing a wide range of mineral ions from groups I to IV and rare earth elements [39]. Bio- and chemiluminescent reactions [40] are examples of reactions occurring in the dark and producing light. These reactions involve complex systems involving condensed heteroaromatic rings, molecular oxygen or hydrogen peroxide, potassium hydroxide and complexed Fe^3+^/Fe^2+^ ions participating in redox reactions. The SBP investigated in the present work, derived from the MBW anaerobic digestate, and the compost-derived SBPs all have the organic and mineral components [1] for chemical reactions to occur in water in the dark. 

Thanks to their complex chemical composition, compost-derived SBPs have been shown to perform as photosensitisers favouring photo-Fenton processes under irradiation with simulated solar light [1,26]. The Fe ions in SBPs are inherited from the pristine MBW from which the products are obtained. These ions are bonded to the organic matter of SBPs. In this fashion, SBPs can maintain Fe(III) in solutions at pH values above 4.5–5. Through this mechanism, SBPs have been demonstrated to induce the oxidation of organic pollutants in water with significant mineralisation of organic C and/or their self-oxidation [26]. 

The autocatalytic properties of the SBP obtained from the MBW anaerobic digestate, which are disclosed in the present work, are a novelty and are valuable for many aspects. The results described in Section 2.1, Section 2.2 and Section 2.3 offer scope for further research aiming to increase the scientific knowledge of advanced oxidation processes [24,25], biomass-derived C catalysts [28] and photo-assisted chemical reactions under light and dark conditions [39,40]. At the same time, they open new perspectives for developing the consumption of no energy and no reagents and clean, mild oxidation processes for the valorisation of MBW as feedstock for the production of new value-added chemical specialities. 

### 3.2. Criticalities for the Implementation of the SBP Catalytic Properties

Aside from the scientific value, the results of the present work indicate a number of criticalities that need to be solved to improve and implement the oxidation process for the valorisation of MBW as feedstock for the production of competitive biowaste-based biosurfactants and materials on industrial and commercial levels.

The auto-catalysed reactions in the absence of added H_2_O_2_ (1-D0, 4-L0, 7-D0 and 10-L0) yield the lowest depolymerisation degree of the pristine SBP. This is readily evident in the data in Figure 3 showing the total production of the low molecular weight fractions (R0.2 and P0.2) in the 12–37% range. Unfortunately, the surface tension values of the high molecular weight fractions (Ri, i = 750–20) are too high (γ = 56–69). The treatments in the presence of hydrogen peroxide at the 2–3 H_2_O_2_/C mol/mol ratio give products with slightly better surface tension properties, i.e., γ = 52–53 for the products obtained in the 9-D3 and 12-L3 treatments (Figure 3). However, compared to the treatments in absence of hydrogen peroxide, the degree of depolymerisation induced by the treatments in the presence of hydrogen peroxide is quite high. For example, the production of the R0.2 and P0.2 fractions (Figure 3B) amounts to 75% and 62% for the 9-D3 and 12-L3 treatments, respectively. The treatments at H_2_O_2_/C 0.5–0.1 mole ratios produce also a high degree of depolymerisation of the pristine SBP organic matter, but they yield high molecular weight products exhibiting the best surface activity properties. The surface tension (Figure 5) range from 47 to 49 mN m^−1^ for the R750, R150 and R20 fractions isolated via membrane filtration of the crude soluble products obtained in the treatment at 0.25–0.5 H_2_O_2_/C mol/mol ratios. Furthermore, the R750 and R150 fractions of the crude soluble products obtained from treatments 19-D0.5 and 16-L0.5, respectively, yield the 2 g L^−1^ solutions exhibiting the lightest yellow colours (Appendix A) coupled with the lowest surface tension (Figure 5). Figure 6 shows an example of the colour of the solutions of the R750 fractions obtained from the 7-D0 and 15-L0 treatments and of the R750 and R150 fractions obtained from treatments 19-D0.5 and 16-L0.5, respectively. A lighter whiter colour coupled with a lower surface tension 38 mN/m has been observed only for the R150 fraction isolated from the ozonised SBP [7]. Bleaching and improving surface activity make SBPs more competitive biosurfactants against current commercial products [7,21,22,41,42,43,44].

Considering the induced degree of depolymerisation and the surface tension values of the products, none of the treatments in the presence of H_2_O_2_ seem competitive with the ozonisation of SBP [7] reported to yield 30% of polymers with molecular weights from 100 to ≥750 kDa and biosurfactants’ solutions with a white colour and surface tension values of 38 mN/m at the product critical micellar concentration of 0.47 g/L. Compared to the ozonisation of SBP, the auto-catalysed reactions in the absence of added H_2_O_2_ (1-D0, 4-L0, 7-D0 and 10-L0) are the only ones producing a definite improvement, although only for the reduced degree of depolymerisation and not for the products’ surfactants properties. 

Undoubtedly, obtaining valuable products by just keeping the SBP in a water solution is the most desirable lowest cost process, as it would not involve the consumption of energy and reagents and process wastes needing secondary treatments for their disposal. The results obtained via the SBP treatments in the absence and presence of hydrogen peroxide at 0.1–0.5 H_2_O_2_/C mol/mol ratios offer scope for investigating the reaction of SBP with water containing hydrogen peroxide in catalytic amounts at the H_2_O_2_/C μmol/mol level to perform as a precursor of active O and OH radicals. Such a system might allow for controlling the rate of the SBP chemical oxidation better in order reduce the degree of depolymerisation and improve the surface activity properties of the products, compared to the products obtained in the present work.

### 3.3. Scopes and Perspectives of Future Improvements

The underlying concept of the previous work [1] was the realisation of an MBW-based biorefinery, which could compete with fossil-based refineries for the production of commodities, fine chemicals and chemical specialities [15,21]. The lignocellulose proximates in MBW [2] constitute, in principle, potential polymeric raw materials made of aliphatic and aromatic C types, substituted by a variety of oxygenated functional groups. The mild chemical treatments adopted in the previous and present work aimed to recover the MBW organic matter in soluble form while maintaining as much as possible the functionalised polymeric structure. Contrary to the pristine MBW, the recovered polymeric soluble organic matter could be fractionated into different molecular cuts with different chemical composition, exploiting the different molecular sizes [37] and/or solubilities in acidic and alkaline water [1]. This methodological approach is quite different from the technology used in fossil-based refineries, which encompasses several thermal and/or multistep chemical processes to manufacture organic commodities, fine chemicals and materials [21,45]. 

For the envisioned MBW-based biorefinery to compete with the fossil-based refinery, the previous work addressed several issues, such as the site-specific variability of MBW [27], the process’s sustainability, the products’ replicability with guaranteed specification and performance, and the production flexibility [2], i.e., the capacity of the biorefinery to obtain different products and to modulate their production according to the market demand. Production flexibility is well practiced in fossil-based refineries [46,47,48,49,50]. 

In this context, the development of the autocatalytic SBP process in the presence of hydrogen peroxide at the H_2_O_2_/C μmol/mol level is a further option offered by the results of the present work, which might contribute more economic sustainability and production flexibility to the aimed MBW-based refinery. The improvement in the yield, quality and performance of the surfactants and polymers described in the present work is a highly worthwhile objective to be achieved, upon considering that the current commercial counterpart products account for large shares of the global chemical market. These commercial products are used for the manufacture of consumer chemicals and auxiliaries for industry and plastics, which constitute about 60% of the chemicals sales in the European market [41]. 

## 4. Materials and Methods

### 4.1. Materials and Treatments

The SBP was available from previous work [1]. It was obtained via hydrolysis at pH 13 and 60° from the anaerobic digestate of unsorted municipal food waste. The digestate was supplied by the Italian Acea Pinerolese MBW treatment plant [9] located in Pinerolo (TO). The digestate was hydrolysed in pH 13 water at 60 °C to yield the SBP according to the procedure reported in previous work [1]. The dry product was re-dissolved in plain water at pH 10 and used as a control against the same solution with hydrogen peroxide at 0.1–3 H_2_O_2_ moles per SBP carbon mole added in order to test the effect of the added oxidant in comparison with water as the only terminal oxidant in the control solution. In more details, the SBP (5 g) was dissolved in 150 mL water at pH 10. The solution was kept at room temperature under the following different conditions: in the absence and presence of added hydrogen peroxide at a 0.1–3 H_2_O_2_/C mole ratio, with and without pH control. In the first case, the pH of the solution was found to decrease from 10 to about 5 over 14 days and remained constant afterwards. In the second case, the starting pH 10 was kept constant during the 14 days by adding a 0.2 g L^−1^ KOH solution in the water. The same conditions were applied for the solution kept in a climatic chamber under irradiation with simulated solar light. At the end of the treatments, the solution was centrifuged at 5000 rpm to separate any insoluble material from the soluble phase. The separated insoluble and soluble products were dried at 60 °C. 

### 4.2. Decarbonated CO_2_-Free Samples

The dry soluble products were re-dissolved in water with 5 N HCl added until gas evolution ceased. Afterwards, the solution was centrifuged to separate the precipitated solid from the supernatant liquid phase. The solid was dried first at 60 °C and then in a chemistry lab desiccator over solid NaOH and silica gel. 

### 4.3. Products’ Isolation and Characterisation

The separated soluble phase from the treatments described in Section 4.1 and the decarbonated CO_2_-free samples were further processed via sequential membrane ultrafiltration through 8 polysulphone membranes with decreasing molecular cutoffs to collect the retentates at 750 kDa (R750), 150 kDa (R150), 100 kDa (R100), 50 kDa (R50), 20 kDa (R20), 5 kDa (R5), 0.2 kDa (R0.2) and the final permeate at 0.2 kDa (P0.2). The obtained retentate and permeate fractions were dried at 60 °C to a constant weight and analysed for their volatile solids, ash, C, and N content. The products were characterised according to their relative C type and functional composition by 13 C solid state NMR spectroscopy and according to their surface tension in water solution with an added product concentration of 2 g L^−1^. 

### 4.4. Analytical Methods

Furthermore, 13C solid state NMR spectra were recorded, as previously reported [1], at 67.9 MHz on a JEOL GSE 270 spectrometer equipped with a Doty probe, available by JEOL (ITALIA) S.p.A. located in Basiglio (MI), Italy. The cross-polarisation magic angle spinning (CPMAS) technique was employed, and for each spectrum, about 104 free induction decays were accumulated. The pulse repetition rate was set at 0.5 s and the contact time at 1 ms, the sweep width was 35 KHz and MAS was performed at 5 kHz. Under these conditions, the NMR technique provides quantitative integration values in the different spectral regions. Thus, the relative composition of C types and functional groups for each product in Figure 1 is based on the integration of the band areas in the 13 C NMR spectrum falling in the following chemical shift (δ, ppm) ranges: 0–53 for aliphatic (Af) C, 53–63 ppm for amine (NR) and methoxy (OMe) C, 63–95 ppm for alkoxy (OR) C, 95–110 ppm for anomeric (OCO) C, 110–160 ppm for total aromatic (Ph) C and 160–185 ppm for carboxylic and amide (COX, X = OR, OM, NR, R = H, alkyl and/or aryl) C. The total integrated bands area was assumed to represent the total C moles in the analysed sample. All other analytical and product characterisation details were as previously reported [1]. 

## 5. Conclusions

This work shows that the realisation of the long-term vision to convert municipal biowaste treatment facilities into cost-effective biorefineries producing biofuel, and value-added biosurfactants and biopolymers for the manufacture of consumer products and plastic materials, requires improvements in process yields and the products’ quality, relative to the current state of the research so far carried out [2]. Process improvements should focus on increasing the yield of the high-molecular-weight water-soluble products by reducing the depolymerisation and/or mineralisation of the pristine material. Product improvements should focus on enhancing the products’ surfactant abilities, workability and mechanical properties. 

The data collected in the present work show that, due to the complex chemical composition of the feedstock natural waste materials and the derived products, it is not easy to assess the trade-off between destroying potentially valuable organic matter and gaining value from the residual matter surviving the chemical treatment. Considering the stakes depicted in Section 1.2, further research on reducing depolymerisation and improving the production yield and properties of the SBP high molecular weight fractions is highly worthwhile. A milder oxidation system, comprising an SBP as a substrate and catalyser, hydrogen peroxide in catalytic amounts (e.g., at H_2_O_2_ μmoles per SBP C mole) to perform as a precursor of active O and OH radicals, and water as a solvent and terminal oxidant, may allow one to better control the rate of the SBP chemical oxidation, reducing the degree of depolymerisation and improving the surface activity properties of the products, compared to the products obtained in the present work.

## Figures and Tables

**Figure 1 molecules-29-00485-f001:**
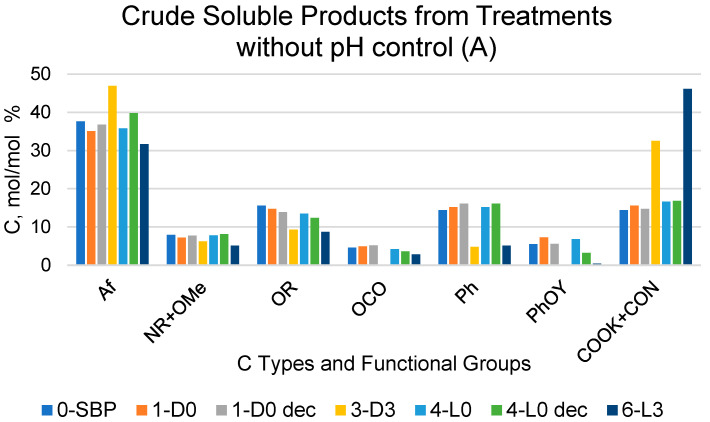
Relative composition of C types and functional groups as mol/mol %, relative to total organic C for crude soluble products, with (**A**,**B**) obtained according to the treatments identified by the abbreviations in Table 1 and for their fractions (Ri, i = 750, 150) with molecular weights ≥ 150 kDa and (**C**) obtained via sequential membrane ultrafiltration (see Section 4). The suffix “dec” for the samples in the subfigures A and B stands for decarbonated samples treated with HCl, i.e., (**A**) 1-D0 dec and 4-L0 dec; (**B**) 7-D0 dec, 10-L0 dec and 11-L2 dec.

**Figure 2 molecules-29-00485-f002:**
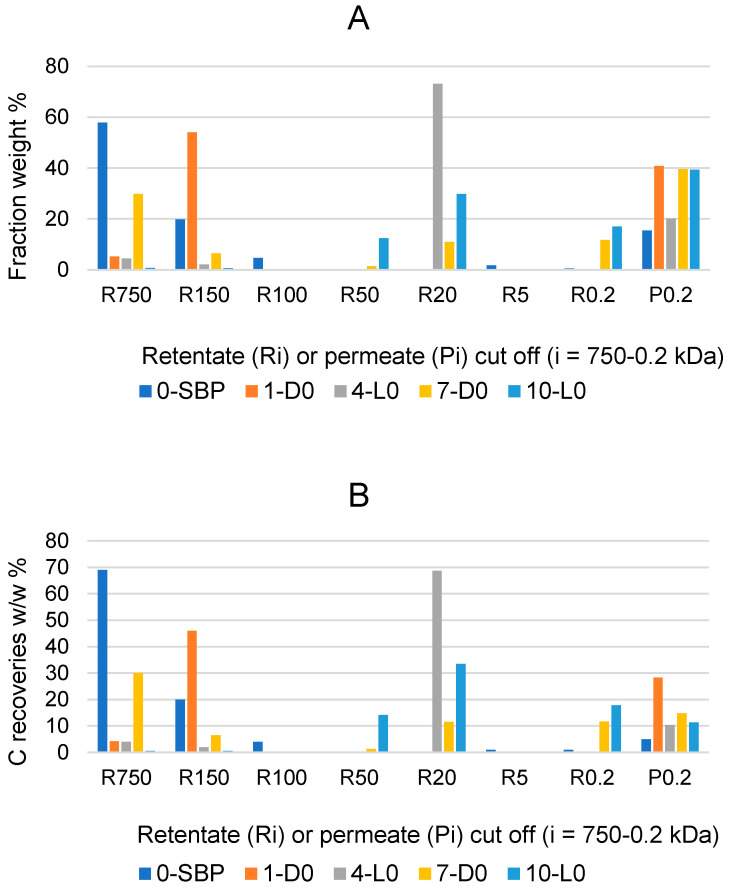
Weight % composition (**A**), and carbon recoveries (**B**) relative to pristine SBP for fractions isolated via sequential filtration through the membranes with decreasing molecular cutoffs at 750, 150, 100, 50, 20, 5, and 0.2 kDa. Figure legends: Ri (i = 750 through 0.2); P0.2 is the permeate though the 0.2 kDa cutoff membrane. Colours correspond to different crude soluble materials identified according to the treatment No. and type in Table 1, i.e., 0-SBP for the pristine SBP; 1-D0, 4-L0; 7-D0; 10-L0 for the products recovered in the treatments of SBP in absence of added H_2_O_2_.

**Figure 3 molecules-29-00485-f003:**
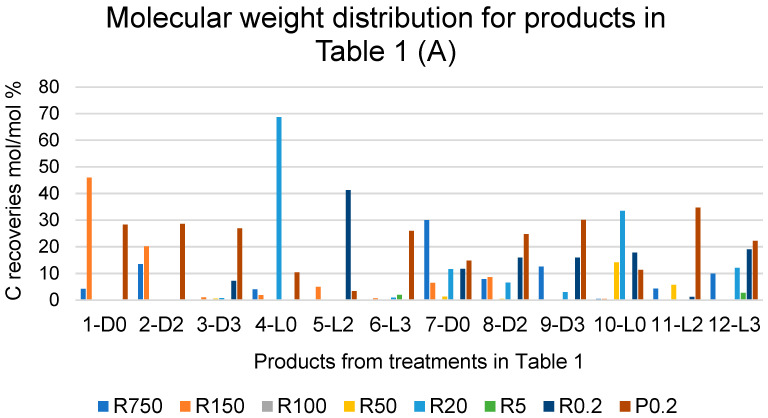
(**A**) Breakdown of C recoveries with fractions obtained via sequential filtration through the membranes with decreasing molecular cutoffs at 750, 150, 100, 50, 20, 5, and 0.2 kDa (see legends in Figure 2) of crude soluble products in all treatments listed in Table 1. (**B**) Total C recovered with the retentate (R0.2) and permeate (P0.2) fractions as % of total C recovered with all fractions. See also Appendix A.

**Figure 4 molecules-29-00485-f004:**
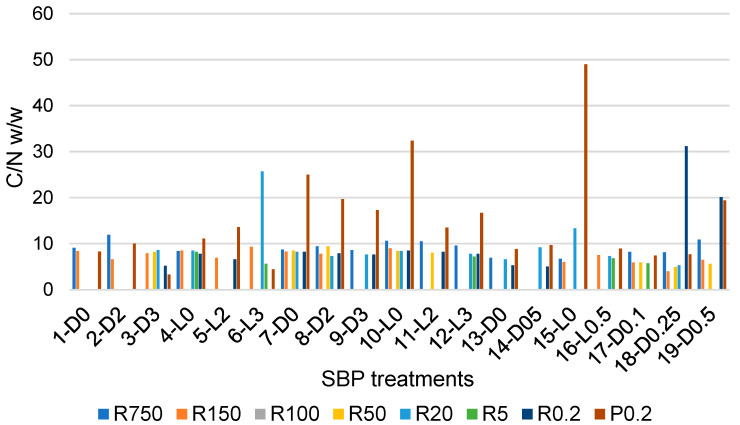
C/N ratio in retentates (Ri, i = 0.2–750) and final permeate (P0.2) fractions, which were isolated via sequential ultrafiltration through membranes with different cutoffs of crude soluble products obtained in the treatments described in Section 2.1, Section 2.2, Section 2.3 and Section 2.4.

**Figure 5 molecules-29-00485-f005:**
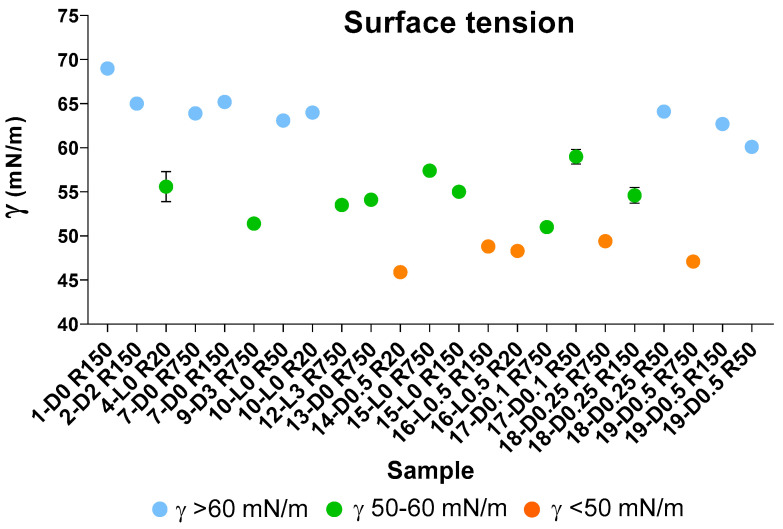
Surface tension values (γ, mN/m) for aqueous solutions containing 2 g/L of retentates (Ri, i = 750–20) isolated via membrane ultrafiltration of crude soluble products obtained in all treatments described in Table 1 and Section 2.1 and Section 2.2. See also Appendix A.

**Figure 6 molecules-29-00485-f006:**
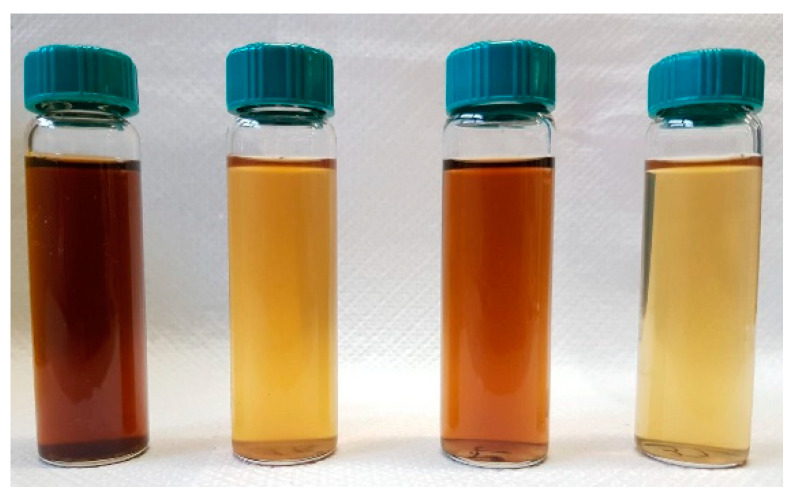
Colour of the solutions of crude soluble products’ fractions, in order from the left side to the right side of the Figure: 7-D0 R750, 19-D0.5 R750, 15-L0 R750, and 16-L0.5 R150 listed in Figure 5 and Appendix A.

**Table 1 molecules-29-00485-t001:** Products obtained from SBP in water solution at 33 g/L concentration, treated ^a^ under different experimental conditions: with and without irradiation by simulated solar light, with and without pH control, in the absence and presence of added hydrogen peroxide at 2–3 H_2_O_2_ moles per starting SBP C mole in the solution.

Treatment ^b^		pH ^c^	KOH ^d^	H^+^/C, ^d^	Recovered Soluble C (mol/mol %) ^e^
No.	Type		grams	eq/mol	
0	none	10			
1	D0	9			79.3
2	D2	5.3			58.2
3	D3	4.9			36.8
4	L0	9			91.8
5	L2	4.9			53.9
6	L3	5.1			35.1
7	D0	10	2	0.24	100
8	D2	10	2.8	0.33	94.1
9	D3	10	3.3	0.40	90.1
10	L0	10	2	0.24	100
11	L2	10	2.9	0.35	84.7
12	L3	10	3.3	0.40	79.3

^a^ See Section 4 for details. ^b^ No. (treatment number), D (dark, no irradiation); L (simulated solar light irradiation); Di and Li, i = 0, 2, 3 H_2_O_2_/C mole ratio. ^c^ Final pH after 14-day treatment; starting pH was 10 at day 0 for all treatments. ^d^ Added KOH (g) to keep pH 10 constant and corresponding acid equivalents produced per pristine organic C mole (H^+^/C eq/mol). ^e^ Recovered soluble C mole %, relative to the C moles in the SBP solution at day 0.

**Table 2 molecules-29-00485-t002:** Mean and standard deviation (StdDev) of recovered soluble C (mol/mol %) for treatments No. 1–6 and No. 7–12.

	Treatments No. 1–6	Treatments No. 7–12
Mean	65.2	99.0
StdDev	20.1	5.9
StdDev%	30.8	5.9

**Table 3 molecules-29-00485-t003:** Mean and standard deviation values calculated for the 1-DO and 1-D0dec samples coupled together and for the 4-L0 and 4-L0dec samples coupled together.

Samples Coupled Together		C Types and Functional Groups						
		Af	NR + OMe	OR	OCO	Ph	PhOY	COX
1-D0, 1-D0dec	Mean	35.9	7.4	14.3	5.1	15.6	6.4	15.1
	Std	1.2	0.35	0.49	0.14	0.64	1.2	0.64
	Std % ^a^	3.3	4.7	3.4	2.8	4.1	18.6	4.2
4-L0, 4-L0dec	Mean	37.8	7.9	12.9	3.9	15.6	5	16.7
	Std	2.8	0.2	0.78	0.42	0.64	2.5	0.14
	Std % ^a^	7.5	2.7	6.0	10.9	4.1	50.9	0.85

^a^ Relative standard deviation as % of the mean values.

**Table 4 molecules-29-00485-t004:** Content of COXorg and CO_2_ carbon (mol/mol %) in samples 1-D0, 4-L0, 7D0 and 10-L0.

Sample	COXorg	CO_2_
1-D0	15.1 ± 0.64	none
4-L0	16.7 ± 0.14	none
7-D0	11.1	25.1
10-L0	12.4	25.0

**Table 5 molecules-29-00485-t005:** Mean C/N value of fractions with the same molecular weight.

Fractions	Mean	StD ^a^	STDr ^b^	Data Points ^c^
R750	8.5	2.7	32	14
R150	7.3	1.4	20	14
R50	7.4	1.5	21	8
R20	9.5	5.2	55	13
R5	6.7	1.1	16	5
R0.2	9.9	7.4	75	13
P0.2	14.4	11.0	76	19

^a^ Standard deviation. ^b^ Standard deviation as % of mean C/N value. ^c^ Number of data points.

**Table 6 molecules-29-00485-t006:** Mean surface tension values (γ, mN m^−1^) for each of the groups with γ < 50, 50 ≤ γ < 60, and γ ≥ 60. See also Appendix A.

	γ ≥ < 60	50 ≤ γ 60	γ < 50
N ^a^	9	9	5
Mean ^b^	64.1 a	54.7 b	47.9 c
Std ^c^	2.4	2.6	1.4

^a^ Number of data points. ^b^ Values with different letters are significantly different at *p <* 0.01 level, e.g., a > b > c. ^c^ Standard deviation.

## Data Availability

All the available data are reported in the present and in the previous referenced publications.

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
