# Peer review of "The Autocatalytic Chemical Reaction of a Soluble Biopolymer Derived from Municipal Biowaste"

_molecules, 2024, doi:10.3390/molecules29020485_

Round 1
Reviewer 1 Report
Comments and Suggestions for Authors
The submitted manuscript (ID: molecules-2786408) explores the behaviour of aqueous solutions containing a soluble biobased polymer (SBP) derived from MBW anaerobic digestate, in terms of its autocatalytic properties. This research aims at validating the future role of SBPs in biorefineries, and particularly, as surfactants.
The scope of the research is of great interest, as it explores the production of value-added products from the anaerobic digestion process by-products (ie. Digestates). The experimental work is solid, and it provides enough information for a clear scientific discussion.
However, I have found some experimental parts of the document unconnected and the discussion a bit difficult to follow due to the great amount of information. Moreover, I miss more references to the research context, to understand why this SBPs could be of relevance in the future of the chemical industry, so as to link with any previous work.
Please, find below some clarifications and suggestions that might be considered for the authors in order to improve the quality of the paper:
0) Abstract:
· - I suggest changing the order of the information shown in the abstract. It starts directly with some experimental conditions of the trials, but the core of the discussion starts by line 15 to 21. A good order could be as follows: a short intro of the paper discussion (line 21-23), followed by the main results described between lines 15-21, and finished by specific experimental conditions of the trials (lines 10-15).
1) Introduction:
· - In line 29 it is mentioned SBPs as soluble biobased products, whereas in the abstract SBP means soluble biobased polymers. Please, correct the terminology along the whole document. It is not clear to me now what concept it is referred in some cases.
· - Please, provide some references to highlight the challenge of the production of SBPs from anaerobic digestate. There are none.
· - In lines 38-39 the composition in terms of organic matter and mineral elements are mentioned, but there is no further explanation regarding that great variety. Please, consider including some additional literature references linking this information to the properties and characteristics of different types of anaerobic digestates from MBW anaerobic digestion.
· - In line 48-50 of the introduction, couple drawbacks related to the use of SBPs in detergents and/or plastic industries are detailed, mainly related to the presence or aromatic lignin-like moieties. Please, provide references.
· - Line 56-57: I miss some clear context to explain why the autocatalytic properties of SBPs are relevant. There is some references to different applications, but it might not be sufficiently clear. Please, provide some clarifications and references, if needed.
· - Line 71-72: what is the purpose of describing this behaviour? This is related to my previous comment. I believe it is the biorefinery approach, but it should be more clearly detailed.
· - Line 71-72: the valorisation of anaerobic digestates to different value-added products than soil amendments/composts is quite innovative. I suggest to the authors giving more emphasis to this topic.
· - Line 77: it is mentioned that sourcing MBW chemical compositions is not well known. I would rather mention that monitoring the MBW composition is challenging due to its heterogeneity (it does not mean the same, under my point of view). I suggest to re-write this sentence.
· - Line 84-85: indeed, you are demonstrating and validating the protocol, right? Please, provide some additional emphasis to this.
2) I suggest moving to section 2 the information mentioned in section 4 “Materials and Methods”.
3) Results:
· - Lines 96-97: Is there any previous experience with MBW to complement the previous references, together with lignocellulosic materials? If so, please include details.
· - Line 98-99: for me, this information (the aim of the work) should be placed in the abstract and/or the introduction section.
· - Line 426-429: is this a hypothesis, or is there any previous work that could support this information? If so, please, include reference.
· - Line 446-447: I guess this tests are performed in order to assess the products performance as surfactants? Please, provide some context.
· - Lines 594-613: this is not a conclusion. It is mainly state of the art. Should be moved to introduction section.
· - Lines 630-644: it is not a conclusion from the worked presented previously. It is not linked with any previous information. It should be removed or moved to introduction section
4) Materials and methods:
· - In general, I miss some explanation related to the reasons why you are performing those tests. Is it aiming at reproducing any real industrial conditions? Or validating any potential scaled-up process? I suggest to include some mention to this in this section, or in the discussion section.
· - Delete line 666
Author Response
author's notes to reviewer 1 in attached file

Reviewer 2 Report
Comments and Suggestions for Authors
The comments and suggestions for the authors are summarized in the attached file

The English language requires proofreading
Author Response
author's notes to reviewer 2 in attached file

Round 2
Reviewer 2 Report
Comments and Suggestions for Authors
OK